# Decoding Fear: Analysis and Prognosis of Preoperatory Stress Level Through Advanced Statistical Modelling—A Prospective Study Across Multiple Surgical Specialties [note 1]

**DOI:** 10.3390/medsci13030181

**Published:** 2025-09-05

**Authors:** Cristina Gena Dascălu, Andrei Ionut Cucu, Andreea Vovciuc, Sorin Axinte, Serban Turliuc, Amelian Madalin Bobu, Camelia Tamas, Vlad Porumb, Emilia Patrascanu, Catalin Mihai Buzduga, Paula Alexandra Blanaru, Anca Petruta Morosan, Iulian Prutianu, Roxana Covali, Andreea Ioana Pruteanu, Claudia Florida Costea, Alexandru Carauleanu

**Affiliations:** 1Department of Medical Informatics and Biostatistics, Faculty of Medicine, “Grigore T. Popa” University of Medicine and Pharmacy, 16 University Street, 700115 Iasi, Romania; cristina.dascalu@umfiasi.ro; 2Faculty of Medicine and Biological Sciences, University Stefan cel Mare of Suceava, 720229 Suceava, Romania; andreea.vovciuc@student.usv.ro (A.V.); axintesorin@gmail.com (S.A.); 3Suceava County Emergency Clinical Hospital, 720237 Suceava, Romania; 4Faculty of Medicine, University of Medicine and Pharmacy Grigore T. Popa Iasi, 700115 Iasi, Romania; serban_turliuc@yahoo.com (S.T.); amelian.bobu@gmail.com (A.M.B.); camelia6ta@yahoo.com (C.T.); vlad.porumb@umfiasi.ro (V.P.); patrascanu.emilia@umfiasi.ro (E.P.); catalinbuzduga@gmail.com (C.M.B.); blanarupaula@yahoo.com (P.A.B.); morosan.petruta@gmail.com (A.P.M.); pruty04@gmail.com (I.P.); andreea.ioana.dragu@gmail.com (A.I.P.); costea10@yahoo.com (C.F.C.); drcarauleanu@yahoo.com (A.C.); 5Clinical Emergency Hospital St. Spiridon, 700111 Iasi, Romania; 6Faculty of Medical Bioengineering, University of Medicine and Pharmacy Grigore T. Popa Iasi, 700115 Iasi, Romania; ana.covali@umfiasi.ro

**Keywords:** preoperative stress, B-MEPS, surgery, predictors, random forest, anxiety

## Abstract

**Background:** Preoperative stress is a multifactorial phenomenon shaped by physiological, psychological, and social influences, with a substantial impact on postoperative recovery. This study aimed to quantify preoperative stress levels, identify associated factors, and rank their predictive importance. **Methods:** A prospective study was conducted on 197 patients scheduled for general surgery, orthopedics, neurosurgery, or otorhinolaryngology procedures between December 2024 and June 2025 at Suceava County Emergency Clinical Hospital. Stress levels were assessed using the Brief Measure of Emotional Preoperative Stress (B-MEPS), translated and culturally adapted into Romanian. Statistical analyses included nonparametric tests, generalized linear modeling, and Random Forest regression. **Results**: The mean B-MEPS score was 21.42 ± 6.04 (range: 11–34), indicating a moderate level of preoperative stress. Higher stress scores were significantly associated with female sex (*p* < 0.001), lower educational attainment (*p* = 0.003), divorced marital status (*p* = 0.007), a history of cancer (*p* = 0.002), and the type of surgical intervention (*p* = 0.003). Random Forest analysis identified the type of surgery, educational level, and sex as the strongest predictors. **Conclusions:** Preoperative stress is chiefly influenced by the type of surgical procedure, educational level, and sex, with potential synergistic effects among these factors. Early identification of high-risk patients enables targeted, personalized interventions to mitigate anxiety and improve perioperative outcomes. Further research should include formal validation of the Romanian version of B-MEPS and the integration of additional psychosocial variables.

## 1. Introduction

Preoperative stress is a multifaceted phenomenon experienced by patients awaiting surgical intervention, characterized by a combination of physiological, psychological, and behavioral responses. It is considered a reaction to the anticipation of surgery and affects up to 60% of surgical patients worldwide [1,2].

Preoperative stress may manifest in various forms, including anxiety, vulnerability, fear, and distress [3,4,5], as well as through physiological changes aimed at maintaining internal homeostasis. These emotions are primarily triggered by concerns related to the surgical procedure itself, potential pain, and procedural outcomes [5,6,7]. Among the most frequently reported fears are failure to awaken from anesthesia and death [8].

### 1.1. Preoperative Anxiety

Anxiety is a significant contributor to preoperative stress, affecting approximately 48% of surgical patients worldwide [9]. The main factors associated with preoperative anxiety include concerns regarding anesthesia, fear of surgical complications, and apprehension about postoperative pain and disability [9]. Various demographic and clinical characteristics influence anxiety levels, with younger patients, females, and those with limited prior surgical experience reporting higher levels of anxiety [8,9].

Anxiety can induce substantial physiological alterations, including changes in hemodynamic parameters such as increased heart rate and blood pressure, elevated cortisol levels [8,10,11,12], and higher plasma adrenaline concentrations [8]. The surgical stress response may also trigger systemic inflammatory reactions, potentially impairing postoperative recovery and long-term outcomes [13]. Furthermore, evidence suggests that psychological stress can adversely affect immune function and contribute to complications through hormonal pathways, such as dysregulation of cortisol secretion [14,15]. Collectively, these findings underscore the importance of adopting an integrated biopsychosocial approach to perioperative patient care [16].

### 1.2. Factors Influencing Preoperative Stress

To date, studies have identified several categories of factors that may influence preoperative stress: (1) demographic factors, (2) procedural and medical factors, (3) institutional and environmental factors, (4) psychological and emotional factors, and (5) socioeconomic and cultural factors. Among demographic and personal medical history, research has reported associations with advanced age [17,18], female sex [12,19], marital status (being married), and significant family responsibilities [1,20], as well as prior surgical experience [12,21]. Regarding procedural and medical factors, evidence indicates that patients undergoing more complex surgeries, such as cardiovascular procedures [22], and those with comorbidities or chronic illnesses tend to exhibit higher preoperative stress levels [12,23]. Institutional factors, such as the hospital environment [19,24] and interactions with healthcare personnel, particularly in situations of poor or indifferent communication [20,21], have also been shown to affect preoperative stress. Preexisting conditions, including anxiety and depression [12,23], as well as insufficient information regarding the surgical procedure [1,20], represent psychological and emotional contributors that can heighten patient stress prior to surgery. Among socioeconomic and cultural factors, the most significant have been identified as financial concerns [1], cultural beliefs, and the presence of family support systems [19].

### 1.3. Scales for Measuring Preoperative Stress

The relationship between preoperative stress and postoperative outcomes is mediated by physiological responses, coping mechanisms, and psychological resilience, forming a multidimensional construct of critical importance for a comprehensive preoperative assessment of stress [25]. Understanding and quantifying preoperative stress is therefore essential for improving perioperative care and tailoring interventions to mitigate adverse outcomes [4,26].

Over time, various methods have been developed to measure preoperative stress, each with its own advantages and limitations. In addition to well-established physiological measures, such as heart rate variability and blood pressure [27] or electrodermal activity [28], psychometric scales have also been introduced. Currently, the most widely used instruments include the *State Anxiety Inventory* (SAI) [29], the *Brief Measure of Emotional Preoperative Stress* (B-MEPS) [30], the *Amsterdam Preoperative Anxiety and Information Scale* (APAIS) [31], and the *Hospital Anxiety and Depression Scale* (HADS) [32].

Among these psychometric instruments, the B-MEPS scale is considered superior due to its high specificity for the preoperative context, ease of questionnaire administration, and stronger predictive ability in identifying patients at high psychological risk before surgery. Furthermore, B-MEPS offers a comprehensive approach to measurement, demonstrates robust psychometric properties, and can be culturally adapted without compromising its psychometric integrity [33,34]. Schiavo et al. and Caumo et al. emphasized the conciseness and clinical practicality of the B-MEPS compared with longer scales, supporting its use for assessing preoperative stress in surgical patients [3,35].

Key advantages of the B-MEPS include its brief and targeted evaluation of preoperative emotional stress, in contrast to longer instruments such as the SAI or HADS, which assess broader constructs of mood and anxiety disorders [3,36]. Its concise format facilitates rapid screening in clinical settings [35]. Literature evidence has demonstrated moderate to strong correlations between the B-MEPS and other validated scales, supporting its concurrent validity [3,35].

Multiple authors have shown that psychological preparation and coping strategies tailored to stress levels measured by the B-MEPS can improve postoperative pain and recovery [30,37]. Additionally, screening for anxiety and providing targeted support based on B-MEPS scores may enhance both patient outcomes and satisfaction [38].

The present study aimed to assess preoperative stress among patients undergoing general, orthopedic, neurosurgical, and otorhinolaryngology procedures and to identify significant associated factors and their predictive strength. Using multifactorial analysis and a Random Forest model, we sought to rank these factors and develop a practical tool for individualized estimation of preoperative stress risk. Accordingly, our investigation was guided by two key questions: *Who influences preoperative stress?* and *Who predicts it best?*

## 2. Material and Methods

### 2.1. Participant Selection and Measures

This prospective study included 197 patients and was conducted at Suceava County Emergency Clinical Hospital between 1 December 2024 and 1 June 2025, in the Departments of General Surgery, Neurosurgery, Orthopedics, and Otorhinolaryngology. All participants completed the B-MEPS (Brief Measure of Emotional Preoperative Stress) questionnaire [30], previously validated in Brazil and translated into Romanian using a forward-translation method performed by expert linguists. Although a full psychometric validation was not conducted, attention was given to ensuring linguistic and cultural accuracy.

The B-MEPS is a brief psychological assessment tool designed to measure emotional stress related to surgical procedures. It was originally developed in Brazil for use during the preoperative period, particularly to identify patients at increased risk of adverse postoperative outcomes due to psychological stress. The scale was constructed by integrating elements from preexisting instruments that assess emotional stress in the preoperative context through the identification of individual vulnerabilities, including components that evaluate anxiety, depression, future self-perception, and minor psychiatric disorders, thereby providing a comprehensive profile of the patient’s emotional state [3,30]. The B-MEPS has undergone psychometric evaluation to ensure reliability and validity, with a reported Cronbach’s alpha of 0.79, indicating satisfactory internal consistency [3,35]. It classifies preoperative emotional stress into low, intermediate, or high categories. The questionnaire was developed and validated in Brazil and, to the best of our knowledge, has not yet been validated internationally in other countries or languages.

### 2.2. Ethical Considerations

The study was conducted in accordance with the ethical principles outlined in the Declaration of Helsinki and with applicable national regulations governing research involving human subjects. The research protocol was approved by the Ethics Committee of the Suceava County Emergency Clinical Hospital, Romania (approval no. 53/28 November 2024), prior to the initiation of data collection. All participants were informed in advance about the purpose and methodology of the study, as well as the voluntary nature of their participation, and were advised of their right to withdraw at any time without penalty. Written informed consent was obtained from each participant. Questionnaire responses were handled confidentially, and anonymity was ensured throughout the study.

### 2.3. Statistical Analysis

The quantitative variable (overall questionnaire score) was described using the mean, standard deviation, minimum and maximum values, median, and interquartile range, while qualitative variables were presented as absolute and relative frequencies.

In the univariate analysis stage, preoperative stress scores were compared across demographic and clinical patient characteristics using the nonparametric Mann–Whitney and Kruskal–Wallis tests. The choice of nonparametric methods was based on testing score distributions with the Kolmogorov–Smirnov goodness-of-fit test, which indicated deviation from a normal distribution.

For the multivariate analysis, demographic and clinical variables showing statistically significant differences in preoperative stress scores were included in a univariate General Linear Model (GLM) to assess their combined influence. The model incorporated all possible interactions between variables (up to fourth-order interactions). Post hoc comparisons between categories were performed using Tukey’s HSD test, with significance levels adjusted for multiple comparisons. Equality of error variances across groups was evaluated using Levene’s test, which indicated some deviation from homogeneity (*p* < 0.001). However, the model was deemed robust given the sample size and the relatively balanced distribution of patients across the four analyzed surgical intervention categories. The effect size of each variable/factor included in the model was assessed using Partial Eta Squared, interpreted according to Cohen’s guidelines (0.01 ≤ partial η^2^ < 0.06: small effect; 0.06 ≤ partial η^2^ < 0.14: medium effect; partial η^2^ ≥ 0.14: large effect).

In the final stage of the study, the same significant characteristics were entered into a Random Forest regression model to determine their predictive efficiency for preoperative stress scores. The model used a 0.30 proportion for random test data, a 0.50 subsample proportion, and 100 trees, with stopping parameters set at a maximum of 100 nodes over 10 levels, a minimum of 5 child nodes, and at least 5 cases.

In addition to the analyses described, we reported 95% confidence intervals (95% CIs) for all means and between-group differences, alongside effect sizes (rank-biserial r or η^2^) and multiplicity-adjusted *p*-values. For the GLM, we calculated partial η^2^ with 95% CIs. To assess robustness, we performed sensitivity analyses: (i) exclusion of patients with cancer, (ii) stratification by surgical type, (iii) robust/median regression for B-MEPS scores, and (iv) ordinal models for individual items. For the Random Forest, we applied repeated 10-fold cross-validation (100 repetitions), reported RMSE_test with 95% CI, predictor importance with 95% CIs, and inspected calibration plots.

All statistical analyses were conducted using the Statistical Package for the Social Sciences (SPSS), version 29.0 for Windows (IBM Corp., Armonk, NY, USA), and Statistica, version 14 (TIBCO Software Inc., Palo Alto, CA, USA).

## 3. Results

### 3.1. Demographic and Clinical Characteristics

The study cohort had a relatively balanced sex distribution, with a slight predominance of females (59.9%). Age groups were also relatively evenly represented, although the largest proportion of patients (39.1%) were aged between 18 and 40 years. A majority of participants (60.4%) resided in rural areas, and most (79.2%) had completed secondary or higher education. Marital status data showed that most patients were married (65.0%).

Regarding lifestyle habits, 23.9% of patients reported alcohol consumption, 19.8% reported tobacco use, and none reported illicit drug use. Half of the participants (49.2%) had chronic illnesses, and slightly more than half (55.3%) were on long-term medication. In terms of surgical history, 72.6% had undergone previous surgical interventions. Additionally, 10.2% of patients reported a history of cancer.

The patients were divided into four groups of relatively comparable size according to the type of surgical intervention they were scheduled to undergo: general surgery, otorhinolaryngology (ENT) surgery, orthopedics, and neurosurgery. The largest proportion of patients were scheduled for general surgical procedures (28.4%), followed by ENT surgery (26.9%), orthopedics (23.9%), and neurosurgery (20.8%).

Regarding the specific types of procedures performed, most orthopedic surgeries consisted of hip and knee arthroplasties, general surgery procedures were primarily cholecystectomies and appendectomies, neurosurgical procedures were mainly lumbar discectomies for herniated discs, and ENT procedures consisted predominantly of tonsillectomies and adenoidectomies.

The demographic and clinical characteristics of the study cohort are summarized in Table 1.

### 3.2. Results of the B-MEPS Questionnaire

#### 3.2.1. Items 1–4 (Immediate Emotional Symptoms)

Agitation (Item 1): Analysis of patient responses to the B-MEPS questionnaire revealed that the majority reported feeling either very agitated or moderately agitated prior to surgery (71.6%), as well as worried to a considerable degree (63.5%). Nonetheless, nearly half of the patients (48.3%) reported being determined to proceed with the surgical intervention. Indecision (Item 2): Approximately half of the respondents (51.8%) reported feeling moderately or very indecisive, while 31.0% reported no indecision, suggesting a relatively balanced distribution but with a significant proportion notably affected. Worry (Item 3): A total of 63.5% of patients indicated feeling moderately or very worried, reflecting marked concern in the preoperative period. Confusion (Item 4): 39.6% of participants reported moderate or severe confusion, while 60.4% stated they rarely or never experienced this state.

#### 3.2.2. Items 5–8 (Persistent Emotional Reactions)

Sense of Unfulfillment (Item 5): The vast majority of patients (65.0%) reported no sense of unfulfillment, and 76.6% did not feel unhappy. However, a significant proportion (73.1%) acknowledged worrying often or almost always about things that do not matter (Item 6). Excessive Sensitivity to Disappointment (Item 7): 63.9% of respondents frequently experienced this reaction. Tension Related to Recent Concerns (Item 8): Of the 197 patients, 81 (41.1%) almost never experienced tension or restlessness when thinking about recent concerns or interests.

#### 3.2.3. Items 9–10 (Simple Emotional and Somatic States)

Unhappiness (Item 9): Only 23.4% of patients admitted to feeling unhappy, while the majority denied this feeling. Gastric Discomfort (Item 10): More than half of the patients (51.3%) reported experiencing emotional tension manifested as a sensation of discomfort in the stomach.

#### 3.2.4. Items 11–12 (Depressive Profile)

Reaction to Unhappiness (Item 11): 29.9% of patients indicated persistence of a negative state (i.e., they did not cheer up easily). The majority (70.1%) reported that when unhappy, they might appear discouraged but could cheer up without difficulty, considering the state to be temporary. Depressed Mood (Item 12): In 53.3% of cases, external factors were able to alter a depressed mood. Detailed patient responses to the B-MEPS questionnaire are presented in Table 2.

### 3.3. Overall Preoperative Stress Score

Based on patient responses to the B-MEPS questionnaire, an overall preoperative stress score was calculated. For the entire cohort, the mean score was 21.42 ± 6.047, with a median of 21.00 and a range of 11 to 34. Considering that the possible range of scores was 10 to 35, with a central value of 22.50, a first quartile of 16.00, and a third quartile of 29.00, the distribution of responses indicates a moderate level of preoperative stress (Table 3).

The overall mean score was 21.42 (SD 6.05, 95% CI: 20.57–22.27), confirming a moderate level of preoperative stress. By sex, women scored 23.39 (SD 5.56, 95% CI: 22.39–24.39), significantly higher than men, who scored 18.47 (SD 5.56, 95% CI: 17.23–19.71), with a difference of Δ = 4.92 points (95% CI: 3.42–6.42). Significant differences were also found by education: middle school 24.07 (95% CI: 22.62–25.52) vs. high school 20.82 (95% CI: 19.51–22.13) vs. university 20.59 (95% CI: 19.10–22.08). Patients with cancer showed significantly higher stress scores: 25.30 (95% CI: 23.37–27.23) vs. 20.98 (95% CI: 20.09–21.87) in those without cancer.

#### 3.3.1. Preoperative Stress Level by Demographic Data

Statistical analysis revealed that preoperative stress levels were significantly higher in women compared with men (*p* < 0.001), in patients with primary school education compared with those with secondary or university education (*p* = 0.003), and in divorced individuals compared with other marital status categories (*p* = 0.007). Stress scores were nearly identical across the three age groups and slightly higher in patients from rural areas compared with those from urban areas, although these differences did not reach statistical significance.

#### 3.3.2. Preoperative Stress Level by Medical History

Similarly, stress scores were comparable between patients who consumed alcohol and those who did not, between smokers and nonsmokers, between patients with and without chronic diseases, and between those with and without a history of previous surgical interventions. Scores were slightly higher in patients receiving chronic medication compared with those who were not, but the difference was not statistically significant. In contrast, preoperative stress levels were significantly higher in patients with a history of cancer compared with those without (*p* = 0.002).

#### 3.3.3. Preoperative Stress Level by Type of Surgical Intervention

Regarding the type of surgical procedure, statistically significant differences were observed in stress scores according to the surgical specialty (*p* = 0.003). The highest values were recorded in patients scheduled for orthopedic procedures—predominantly major operations such as hip and knee arthroplasty, which constituted the majority of the orthopedic group—while the lowest values were seen in patients scheduled for ENT surgery. Patients scheduled for general surgery or neurosurgical procedures had relatively similar and lower stress scores compared with the orthopedic group (Table 3 and Figure 1).

We identified three demographic criteria (sex, educational level, and marital status) and one clinical criterion (history of cancer) that were associated with statistically significant differences in preoperative stress levels, along with the type of surgical intervention for which patients were scheduled. Based on these results, in the second stage of the study, all these categorical variables were included in a univariate General Linear Model (GLM) to investigate whether they exert a combined influence on preoperative stress. The overall model was statistically significant (F (48, 148) = 8.344, *p* < 0.001), with a coefficient of determination R-squared = 0.730 (Adjusted R-squared = 0.643), indicating that the variables included in the model explained 73.0% of the variation in preoperative stress scores (Table 4).

All four criteria included in the model, including the surgical specialty (general surgery, orthopedics, neurosurgery, or ENT), considered the primary factor and had statistically significant effects on preoperative stress scores. In addition, several statistically significant interactions were identified, including those between patient sex and educational level (*p* = 0.003), between educational level and marital status (*p* = 0.001), and between educational level and history of cancer (*p* < 0.001). Furthermore, the type of surgical intervention combined with each possible pair among the three selected demographic criteria showed statistically significant effects on preoperative stress scores.

### 3.4. Identification of Predictive Factors for Preoperative Stress

A Random Forest analysis was performed to rank the five identified predictors according to their potential to estimate preoperative stress scores and to generate actual predictions of these scores.

The initial Random Forest model (five predictors) yielded a RMSE_test ≈ 5.5, with a 95% CI of 4.7–6.3, indicating a relatively large prediction error given the score scale (10–35). Reconstructing the model with only three predictors (surgery type, education, and sex) reduced the test error to RMSE_test ≈ 4.9 (95% CI: 4.2–5.6) and diminished overfitting, while preserving the same predictor ranking. Median relative importance (95% CI) was as follows: surgery type 1.00 (0.92–1.00), education 0.94 (0.83–0.99), sex 0.77 (0.63–0.88).

The overall model assessment (Figure 2A) revealed a rapid decrease in the mean squared error (MSE) for both the training and test datasets, with stabilization after approximately 30 trees were constructed. For the training data, MSE stabilized around 20.0 (19.61 ± 2.227), while for the test data, it reached a plateau around 30.0 (30.29 ± 4.875), which was considerably higher.

This indicates that although the model learns reasonably well from the training data, there is evidence of overfitting, meaning that it does not generalize accurately to new data. Moreover, ${RMSE}_{test}=\sqrt{{MSE}_{test}}=\sqrt{30.29}\approx 5.5$ suggests that the model’s predictions for preoperative stress scores differ from the actual values by ±5.5 units. This error is relatively large given that the score range is relatively narrow (10–35). The most influential predictor for estimating the preoperative stress score was the type of surgical intervention, which was used as the reference variable (Figure 2B). Other highly influential predictors, with importance values above 0.9, were educational level and patient sex. In contrast, marital status showed only moderate influence, while a history of cancer was found to have a weak effect. The decision tree generated by the model began with patient sex as the primary differentiating variable, with female patients tending to have higher preoperative stress scores compared with male patients (Figure 3).

Marital status influenced preoperative stress scores only among male patients, with married or widowed men reporting lower scores compared with unmarried or divorced men. Regarding educational level, women with high school education had higher stress scores than those with university degrees. The type of surgical intervention significantly affected stress scores among women, with the highest scores observed in neurosurgical procedures and the lowest in ENT surgery.

For women with secondary education, neurosurgical procedures were perceived as equally stressful as general surgery, with lower scores recorded for orthopedic and ENT interventions. Conversely, for women with university or primary education, neurosurgical procedures were considered as stressful as orthopedic surgeries, while lower scores were noted for general surgery and ENT interventions.

The correlation analysis between observed and predicted values in the training set indicated that, although the model generated reasonably accurate predictions, there remained substantial variability compared with actual scores. The data points were notably scattered around the ideal reference line y = x, and the correlation between actual and predicted values was below 0.6. This reflects instances where higher actual scores were underestimated by the model and lower actual scores were overestimated (Figure 4A).

A similar pattern was observed in the test dataset, where the correlation between actual and predicted values was even lower. The model tended to underestimate actual scores above 28, avoiding extreme values and pulling predictions toward the mean, a bias typically seen with imbalanced data (Figure 4B).

Given these shortcomings, we reconstructed the model using only the three most important predictive factors in order to improve its performance: type of surgical intervention, educational level, and patient sex. Once again, the overall model analysis showed a rapid decrease in mean squared error (MSE) for both the training and test datasets, with stabilization occurring after the construction of approximately 20 trees, fewer than in the initial model (Figure 5A).

For the training dataset, the MSE stabilized at approximately 21.0 (21.24 ± 2.423), while for the test dataset, it stabilized at approximately 24.0 (24.07 ± 4.081), which is relatively close. Although the error for the training data increased slightly, the error for the test data decreased substantially, indicating that the model is more balanced and that the overfitting phenomenon has been reduced. In this case, ${RMSE}_{test}=\sqrt{{MSE}_{test}}=\sqrt{24.07}\approx 4.9$, which means that the model’s predictions for preoperative stress scores differ from the actual values by ±4.9 units—an error that is smaller and can be considered acceptable. The most influential predictor for estimating preoperative stress scores remained the type of surgical intervention, used as the reference variable (Figure 5B). Educational level also remained a highly important predictor, with an influence score above 0.9, whereas the contribution of patient sex decreased slightly (0.767) but remained significant. In this improved model, the decision tree began with the type of surgical intervention as the primary differentiating variable (Figure 6).

We observed that patients undergoing neurosurgical or orthopedic procedures had higher preoperative stress scores than those scheduled for general surgery or ENT procedures. At the next level, patient sex emerged as an influential factor, with women tending to have higher stress scores than men, regardless of the type of intervention. At the lowest level, educational level influenced stress scores, but with differing trends: women with a higher educational level showed higher stress scores, whereas among men, the highest scores were observed in those with only primary education. The highest mean preoperative stress score was recorded in women with secondary (high school) education who underwent general surgery (M = 27.20), whereas the lowest score was observed in men undergoing general surgery or ENT procedures, regardless of educational level (M = 15.71) (Figure 6). The correlation analysis between observed and predicted values in the training set again demonstrated substantial variability from the actual scores, with points scattered around the ideal *y* = *x* reference line. Higher actual values tended to be underestimated by the model, while lower actual values tended to be overestimated (Figure 7A).

The model exhibited a marked tendency toward regression to the mean, with difficulty in learning the extreme variations of preoperative stress scores. The considerable vertical dispersion of points (residual variation) indicated potential prediction errors. The same phenomenon was observed in the test dataset (Figure 7B). Although some observations were located near the ideal line (*y* = *x*), most were scattered around it. A systematic underestimation of higher values and overestimation of lower values was evident, partially reflecting the limitations of the model. Nevertheless, the results showed that the model built using the three selected predictors, out of the five initially identified, was still more effective in predicting preoperative stress scores.

## 4. Discussion

### 4.1. Influence of Sex on Preoperative Stress

Numerous studies in the literature have reported that women exhibit higher levels of preoperative stress or anxiety, as measured by the B-MEPS or other comparable scales, with significant implications for perioperative management [14,15,30,35,37,39,40,41,42,43]. Some authors have suggested that, in addition to elevated anxiety, women may also display greater vulnerability to emotional stress in the preoperative setting [15]. Evidence indicates that women not only experience more anxiety than men but may also have different coping needs, potentially influencing postoperative outcomes in distinct ways [15]. Consistent with these findings, our study also observed higher stress levels in women compared with men, a phenomenon well-documented in the literature, where it is often linked to differences in pain perception, emotional responses to medical procedures, and the higher prevalence of anxiety disorders among women [15,38].

Women often report higher levels of anxiety related to anesthesia and the surgical procedure itself, with common fears including failure to regain consciousness and experiencing postoperative pain [42,44]. Additionally, women may experience stress related to family responsibilities, which can exacerbate preoperative anxiety levels [45]. In this regard, in a study involving 231 patients scheduled for surgery, Khalili et al. reported that gender norms and expectations may also play an important role, as women may feel additional pressure to conform to societal standards, thereby increasing their anxiety levels [46]. Another explanation could be related to the different coping mechanisms employed by women, who tend to use emotion-focused strategies that may not always effectively reduce anxiety levels [45].

Furthermore, women generally have a greater desire for detailed information about their surgical procedure, and when these informational needs are unmet, their anxiety levels tend to increase [47,48]. This suggests that adequate provision of preoperative education could help mitigate anxiety. Complementing these observations, animal studies have shown that female rats exhibit higher corticosterone levels under preoperative stress conditions compared with males, with females demonstrating an even more pronounced adrenocortical response to stress [49].

### 4.2. Influence of Educational Level on Preoperative Stress

In our study, educational level also proved to be an important predictor, as patients with only primary education had higher preoperative stress scores compared with those with secondary or university education. This may be explained by limited access to medical information and, consequently, a reduced ability to understand procedures and associated risks, which can amplify uncertainty and anxiety. A higher educational level provides a better understanding of medical phenomena and actual associated risks.

In their study, Jones et al. observed that patients with low health literacy often reported not receiving sufficient information about surgery or finding the information provided by physicians difficult to understand, both of which contributed to increased anxiety levels [50]. Similarly, Okediji et al., in a study of 100 randomly selected preoperative participants from several hospitals, reported that patients with lower educational levels exhibited greater preoperative anxiety compared with those with higher education, emphasizing the role of education in managing preoperative stress [51]. Anxiety arising from an inadequate understanding of surgical procedures can also trigger physiological stress responses, such as elevated blood pressure, which may complicate both surgery and postoperative recovery [52]. In such situations, Jones et al. recommend the use of visual aids and simplified educational materials, noting that these are often underutilized yet could help bridge the gap for patients with lower educational levels [50].

Ambrose and Atzeni likewise concluded that effective preoperative education can mitigate these effects by reducing anxiety and mentally preparing patients for surgery [53]. Providing patients with adequate information reduces anxiety and meets their strong desire for knowledge about the surgical procedure and anesthesia [2,20,54,55]. Studies have highlighted that greater information needs are associated with higher anxiety levels, and educational strategies should be consistently applied to alleviate patient distress [2,54], while taking into account cultural and educational factors [54,56].

### 4.3. Influence of Marital Status on Preoperative Stress

Regarding the influence of marital status on preoperative stress, we found that the highest levels were observed among divorced patients, while the lowest were reported by widowed individuals. Married and single patients showed relatively similar and intermediate stress scores between these two extremes. These findings are consistent with the literature, where research has indicated that single individuals tend to experience higher levels of perceived stress compared with married individuals. Such stress is often related to social commitments, loneliness, and financial concerns, which can exacerbate anxiety levels in the preoperative period [57]. Conversely, married individuals may benefit from stronger support systems, which can mitigate anxiety. Ching and Annisa demonstrated that family support significantly reduces anxiety in preoperative patients, suggesting that the presence of a spouse or family members may provide emotional stability [58].

Moreover, studies have shown that divorced or separated individuals are at increased risk of major adverse cardiovascular events due to heightened neural–immune mechanisms associated with stress [59]. In contrast, among patients with chronic illnesses such as cancer or cardiovascular disease, marital status does not appear to significantly influence anxiety levels, although it may impact other aspects of psychological distress, such as depression [60].

In the case of divorced individuals, personalized counseling sessions that address marital status-specific factors such as loneliness or financial concerns may be beneficial, as they can help single patients develop coping strategies to manage their anxiety [61].

Interestingly, although both divorced and widowed individuals are single, in our study, preoperative stress levels were perceived differently: widowed patients reported lower levels compared with divorced patients. Widowed individuals, particularly women, often experience a significant reduction in stress over time compared with their married counterparts. This may be largely due to a decrease in household production activities, as widows tend to engage in such activities less frequently [62]. Moreover, studies have shown a slight increase in the time widows spend with friends and relatives, which can provide emotional support and reduce stress [62].

Additionally, widowed individuals often report an increase in religious or spiritual beliefs following the loss of a partner, which has been associated with reduced grief and stress. This increase in religiosity can serve as a coping mechanism, providing emotional compensation and lowering stress levels [63]. Although widowhood is linked to higher rates of depression, widows often report lower levels of anxiety and stress [64].

### 4.4. Influence of Cancer Diagnosis on Preoperative Stress

Patients with cancer often experience elevated levels of anxiety and stress prior to surgery, attributable not only to the imminence of the surgical intervention itself but also to the cancer diagnosis, which represents a major life stressor. In this regard, several studies have reported that 32.5% of cancer patients exhibited preoperative anxiety and 16.11% presented with preoperative depression [65]. Furthermore, the type of surgical intervention significantly influences preoperative stress levels [65], with patients diagnosed with pancreatic cancer experiencing particularly pronounced preoperative anxiety due to the lethal nature of the disease and the complexity of the surgical procedure [66].

Our findings similarly revealed that patients with a history of cancer had statistically significantly higher preoperative stress levels compared to other patients, an observation that was expected, given the substantial psychological impact of such a diagnosis and the inevitable uncertainties surrounding the surgical outcome and prognosis. Preoperative stress not only has psychological consequences but also induces metabolic disturbances and immune system alterations, which may increase the risk of postoperative complications [67,68].

In a recent animal study, in which breast and colon cancer cell lines were directly injected to establish experimental models of metastasis, Matzner et al. demonstrated that stressful preoperative periods suppressed immune responses, such as reduced plasma IL-12 levels, which may impair resistance to tumor metastases [68].

### 4.5. Influence of Surgical Procedure Type on Preoperative Stress

Research focusing on populations such as cancer patients, elderly individuals undergoing joint replacement, patients undergoing cardiac surgery, and those undergoing outpatient hand surgery has revealed variations in anxiety profiles and relevant predictors [25,65,69,70,71].

Several original studies have reported that the type and invasiveness of surgery influence patients’ preoperative stress, with more complex and higher-risk procedures being associated with greater stress levels. Cardiac surgeries [72], aortic or peripheral vascular procedures [22], and major abdominal surgeries [73] tend to induce higher preoperative stress due to the perceived risk and potential for complications.

Our results confirm the working hypothesis that patients’ preoperative stress levels are significantly influenced by certain demographic and clinical factors, the most important of which is the type of surgical procedure. The highest preoperative stress scores were observed in orthopedic surgeries, while the lowest were recorded in ENT procedures. Neurosurgical and general surgical interventions had relatively similar and comparatively low preoperative stress scores. These findings can be explained by the fact that the most complex surgeries in our cohort were orthopedic procedures, predominantly hip and knee arthroplasties. By contrast, the most common neurosurgical operations were surgical treatment of lumbar disc herniation, the most frequent general surgical procedures were appendectomies and cholecystectomies, and in ENT surgery the most common interventions were tonsillectomies and adenoidectomies. Therefore, in the context of this study, it appears that it is not the surgical specialty per se (general surgery, orthopedics, neurosurgery, or ENT) that correlates with preoperative stress, but rather the specific type of surgical intervention itself.

Orthopedic surgeries, particularly knee and hip arthroplasties, are generally more complex and have longer operative times, resulting in increased physiological stress for both patients and surgeons [74,75]. These observations are also supported by our findings, as the most complex surgical procedures in our cohort were orthopedic.

In a study evaluating morning anxiety prior to various surgical procedures in 186 patients with ENT pathology, Binar et al. reported that patients in the head and neck surgery group exhibited higher levels of preoperative anxiety compared to other groups, particularly those undergoing rhinologic and otologic procedures [76]. Similar findings have been reported in studies assessing preoperative stress in patients with abdominal pathologies: Duymus et al. found that patients with benign tumor pathologies had lower levels of anxiety compared to those with cancer [77]. Furthermore, in a study comparing preoperative stress in women with breast cancer scheduled for mastectomy versus patients with cholecystitis scheduled for cholecystectomy, Ghanbari et al. observed significantly higher psychological preoperative stress in the cancer group compared to the benign pathology group [78]. A similar pattern is observed in neurosurgical procedures: studies have shown that the prevalence of preoperative anxiety in patients with brain tumors can be as high as 89% [79], whereas in patients with less severe pathologies, such as lumbar disc surgery, the prevalence of preoperative anxiety is lower, approximately 43% [80].

Patients with prior exposure to anesthesia or surgery tend to have lower levels of anxiety, suggesting that familiarity with the surgical process may help mitigate stress [9].

### 4.6. Identification of Predictors of Preoperative Stress

Several studies in the literature have attempted to identify predictors of preoperative stress. Xing et al. identified significant predictive factors such as fear of the surgical procedure, quality of sleep before surgery, and perceived social support [81]. Similarly, in a study of 231 patients, logistic regression analysis by Khalil et al. found that age, female gender, and awareness of potential anesthetic side effects were the most predictive factors for situational anxiety, whereas for trait anxiety, the strongest predictors were age, female gender, and place of residence [46].

In another study conducted on 239 patients scheduled for surgery, Nigussie et al. reported that the most important predictors of preoperative anxiety were trait anxiety, marital status (being single or divorced), timing of surgery, and income level [82]. Likewise, Woldegerima et al. identified age over 60 years, emergency surgery, preoperative pain, and rural residence as significant predictive factors [83].

Our study results indicate that the level of preoperative stress depends not only on patients’ individual characteristics but also on the way in which these characteristics interact with one another. The factors with the highest predictive power for preoperative stress were found to be (1) the type of surgical procedure, the most important predictor in both Random Forest models and with a large effect size in multifactorial analysis; (2) educational level, with a very strong influence (>0.9 in Random Forest) and a large effect in ANOVA; (3) patient sex, an important predictor with a large effect size in ANOVA; (4) marital status, which showed moderate influence in the initial model and a moderate effect in ANOVA; and (5) history of cancer, which had low influence in Random Forest but a large effect in multifactorial analysis.

Thus, preoperative stress is strongly influenced by the type of surgical procedure, educational level, and sex, with identified interactions suggesting that the effect of one factor may be amplified or attenuated by others. Overall, these findings support the need for a personalized approach to preoperative preparation, one that accounts not only for isolated factors but also for combinations of demographic and clinical characteristics.

The optimized model, built on the three main predictors (type of surgery, educational level, and sex) demonstrated superior generalization capacity and reduced prediction error compared to the initial version, supporting the relevance of these factors as key predictors in estimating an individual patient’s risk of preoperative stress.

Regarding the type of surgical intervention as a predictive factor for preoperative stress, findings in the literature are contradictory: some authors have reported that the type of procedure may be considered a predictor [77,78], while others have found no evidence supporting it as an independent predictor [82,83,84,85]. Even so, the type of surgery may serve as an indirect predictor of preoperative stress, particularly when it involves an oncological diagnosis or major procedures associated with high risk of mortality or loss of function of a vital organ.

### 4.7. Study Limitations

This study has several limitations that should be considered when interpreting the results. Since the B-MEPS scale has not been officially translated and validated in Romania, we cannot claim that our findings represent standardized measures for the Romanian population. Therefore, this work can be regarded as a pilot study for the Romanian context. The inclusion of patients from multiple surgical specialties may have introduced additional variability that is difficult to control statistically. Given that the sample was drawn from a single center, the results may be influenced by characteristics specific to the hospital, medical staff, and local patient population, which may limit the generalizability of the conclusions to other regions or healthcare systems. Another potential limitation is that, despite the reported results, the Random Forest model exhibited difficulties in capturing the extremes of the preoperative stress score distribution, combined with a tendency toward regression to the mean. This may suggest the existence of additional latent variables influencing the stress score that were not included in the present study, such as the level of social support, history of psychological trauma, or degree of trust in medical staff. Because only the Romanian version of the B-MEPS was applied, a Bland–Altman analysis against the original scale was not possible, highlighting the need for future validation studies including both versions. Another limitation is that, although we reported internal consistency and factorial coherence, we did not perform a full psychometric validation (test–retest, factorial invariance, cross-linguistic equivalence). The inclusion of uncertainty estimates (95% CIs) and sensitivity analyses (stratifications, exclusions, robust models, cross-validation for RF) demonstrates that our results are robust, but these do not replace the need for a complete psychometric validation of the Romanian B-MEPS.

## 5. Conclusions

This study demonstrates that preoperative stress is significantly influenced by a set of demographic and clinical factors, with the type of surgical intervention, educational level, and patient sex having the greatest impact. Interactions among these factors may either amplify or attenuate stress levels, underscoring the need for a personalized approach to preoperative preparation. Both multifactorial analysis and the Random Forest model indicated that the type of surgical procedure is the strongest predictor of preoperative stress scores, educational level is inversely correlated with stress, and female sex is associated with higher preoperative stress compared to male sex. Marital status and a history of cancer play a moderate but contextually significant role.

These findings support the need for psychological assessment in the preoperative period, using culturally and linguistically adapted instruments, to identify patients at higher risk. The implementation of personalized interventions, such as psychological counseling, health education tailored to the patient’s level of understanding, or anxiety-reduction strategies, may improve postoperative recovery and enhance patient satisfaction.

Looking ahead, multicenter studies with larger samples and additional psychosocial variables, as well as formal validation of the B-MEPS scale in Romanian, could strengthen the practical applicability of these results and support the development of robust predictive tools for managing preoperative stress in surgical patients.

## Figures and Tables

**Figure 1 medsci-13-00181-f001:**
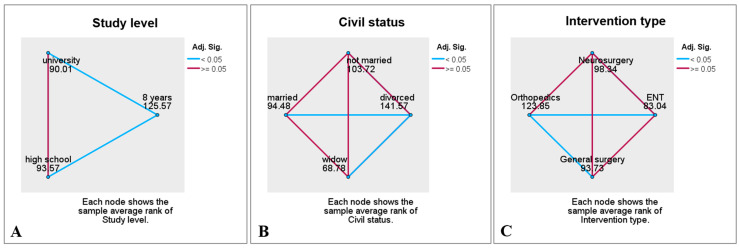
Pairwise comparisons of preoperative stress scores between categories of (**A**) study level, (**B**) civil status, and (**C**) intervention type.

**Figure 2 medsci-13-00181-f002:**
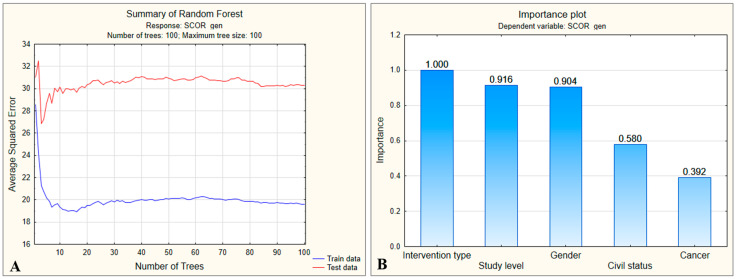
(**A**) Evolution of mean squared error (MSE) with increasing number of trees in the Random Forest model. (**B**) Relative importance of predictors for preoperative stress scores.

**Figure 3 medsci-13-00181-f003:**
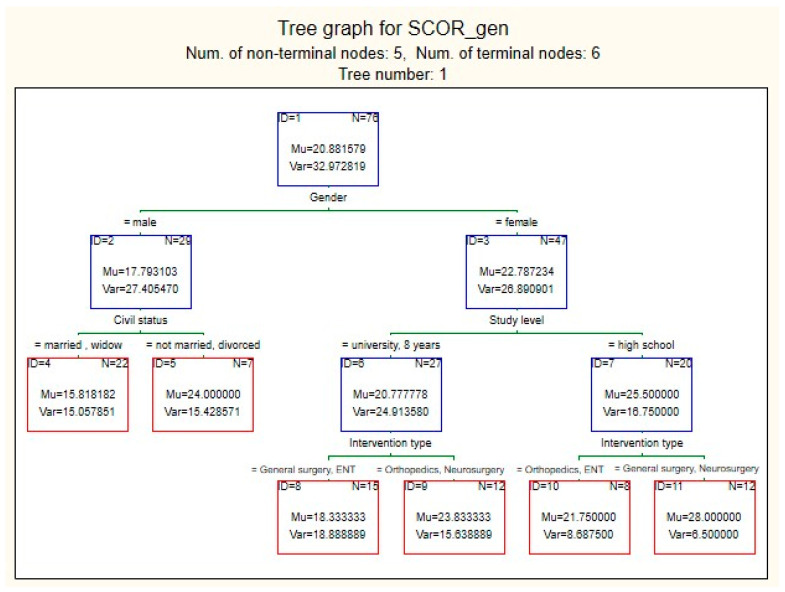
First decision tree from the Random Forest model predicting preoperative stress score (SCOR_gen), *Mu—score average value in node; Var—score variance in node; N—number of cases in node*.

**Figure 4 medsci-13-00181-f004:**
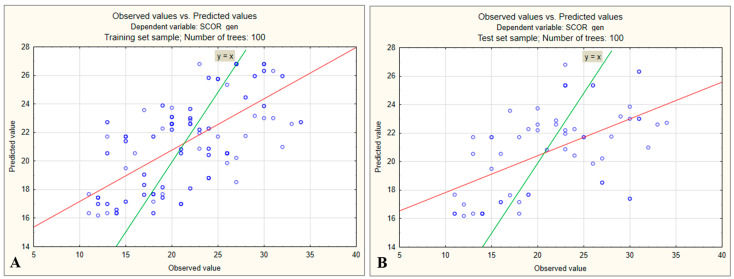
(**A**) Relationship between observed and predicted preoperative stress scores in the training dataset (100 trees). (**B**) Relationship between observed and predicted preoperative stress scores in the test dataset (100 trees).

**Figure 5 medsci-13-00181-f005:**
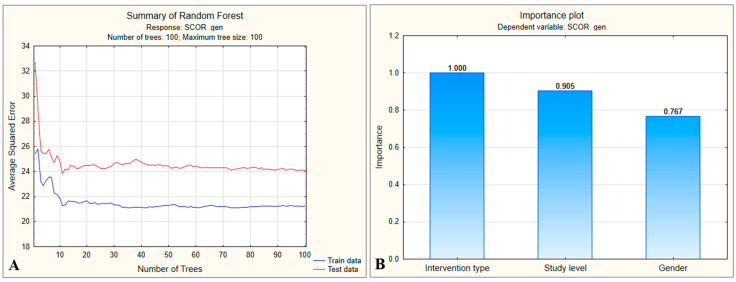
(**A**) Relationship between observed and predicted preoperative stress scores in the training dataset (100 trees). (**B**) Relationship between observed and predicted preoperative stress scores in the test dataset (100 trees).

**Figure 6 medsci-13-00181-f006:**
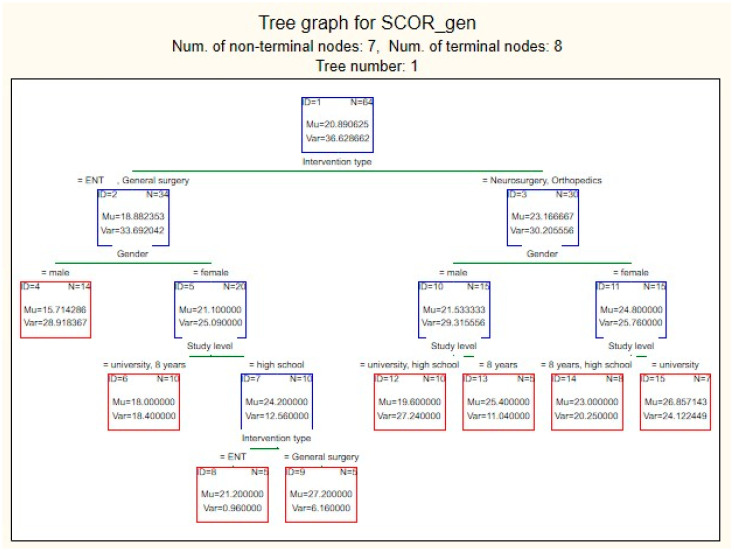
First decision tree illustrating hierarchical splits of predictors for preoperative stress score (SCOR_gen), *Mu—score average value in node; Var—score variance in node; N—number of cases in node*.

**Figure 7 medsci-13-00181-f007:**
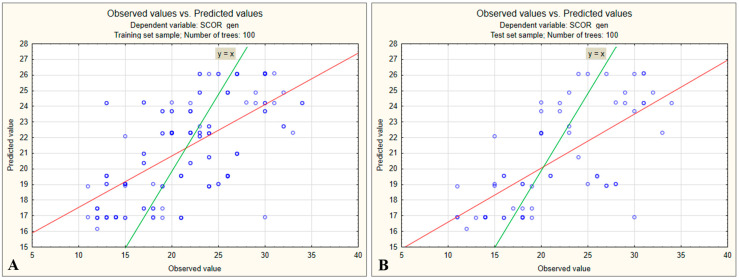
Relationship between observed and predicted preoperative stress scores (SCOR_gen) in the training dataset (**A**) and test dataset (**B**) using the Random Forest model (100 trees).

**Table 1 medsci-13-00181-t001:** Demographic and clinical characteristics of the study cohort (n = 197).

Variables	N (%)	Variables	N (%)	Variables	N (%)
**Gender**		**Marital status**		**Chronic medication**	
Female	118 (59.9%)	Married	128 (65.0%)	Yes	109 (55.3%)
Male	79 (40.1%)	Unmarried	45 (22.8%)	No	88 (44.7%)
		Widowed	9 (4.6%)		
		Divorced	15 (7.6%)		
**Age group**		**Alcohol consumption**		**History of surgical interventions**	
18–40 years	77 (39.1%)	Yes	47 (23.9%)	Yes	143 (72.6%)
41–60 years	58 (29.4%)	No	150 (76.1%)	No	54 (27.4%)
>60 years	62 (31.5%)				
**Place of residence**		**Tobacco use**		**History of cancer**	
Rural	119 (60.4%)	Yes	39 (19.8%)	Yes	20 (10.2%)
Urban	78 (39.6%)	No	158 (80.2%)	No	177 (89.8%)
**Educational level**		**Chronic diseases**		**Type of surgical intervention**	
Primary school	41 (20.8%)	Yes	97 (49.2%)	Orthopedics	47 (23.9%)
High school	88 (44.7%)	No	100 (50.8%)	Neurosurgery	41 (20.8%)
University	68 (34.5%)			General surgery	56 (28.4%)
				ENT surgery	53 (26.9%)

**Table 2 medsci-13-00181-t002:** Detailed patient responses to the B-MEPS questionnaire (n = 197) [30].

**Item & Original B-MEPS Wording**	**Not at All**	**Somewhat**	**Moderately**	**Very Much So**	**Total**	
Item 1: I am jittery	26 (13.2)	30 (15.2)	65 (33.0)	76 (38.6)	197 (100.0)	
Item 2: I feel indecisive	61 (31.0)	34 (17.3)	64 (32.5)	38 (19.3)	197 (100.0)	
Item 3: I feel worried	39 (19.8)	33 (16.8)	47 (23.9)	78 (39.6)	197 (100.0)	
Item 4: I feel confused	66 (33.5)	53 (26.9)	39 (19.8)	39 (19.8)	197 (100.0)	
**Item & Original B-MEPS Wording**	**Almost never**		**Often**	**Almost always**	**Total**	
Item 5: I feel like a failure	128 (65.0)		47 (23.9)	22 (11.2)	197 (100.0)	
Item 6: I worry too much over something that really doesn’t matter	53 (26.9)		83 (42.1)	61 (31.0)	197 (100.0)	
Item 7: I take disappointments so personally that I can’t get them out of my mind	71 (36.0)		82 (41.6)	44 (22.3)	197 (100.0)	
Item 8: I get in a state of tension or turmoil as I think over my recent concerns and interests	81 (41.1)		81 (41.1)	35 (17.8)	197 (100.0)	
**Item & Original B-MEPS Wording**			**No**		**Yes**	**Total**
Item 9: Do you feel unhappy?			151 (76.6)		46 (23.4)	197 (100.0)
Item 10: Do you have a feelings of discomfort in the stomach?			96 (48.7)		101 (51.3)	197 (100.0)
**Item & Original B-MEPS Wording**	**I may look dispirited but I brighten up easily**		**I have pervasive feelings of sadness or feel constantly gloomy**		**Total**	
Item 11: How do you react when you are unhappy?	138 (70.1)		59 (29.9)		197 (100.0)	
**Item & Original B-MEPS Wording**	**Occasional sadness**	**External factors can change it**	**Being without help or hope**		**Total**	
Item 12: How do you describe your depressed mood?	68 (34.5)	105 (53.3)	24 (12.2)		197 (100.0)	

**Table 3 medsci-13-00181-t003:** Comparative analysis of preoperative stress scores according to the demographic and clinical characteristics of the study cohort (n = 197).

Variable/Category	N	Mean	SD	Min	Max	Median	IQR	*p*-Value
**Total**	197	21.42	6.047	11	34	21.00	17.00–26.00	
**Sex**								<0.001 †**
Male	79	18.47	5.560	11	32	18.00	14.00–23.00	
Female	118	23.39	5.555	13	34	23.00	20.00–28.00	
**Age group**								0.883 ‡
18–40 years	77	21.36	6.438	11	34	21.00	15.00–26.00	
41–60 years	58	21.76	5.430	11	33	21.00	18.00–24.50	
>60 years	62	21.16	6.176	12	32	22.00	15.00–27.00	
**Place of residence**								0.407 †
Rural	119	21.74	6.301	11	34	22.00	18.00–26.00	
Urban	78	20.92	5.643	11	30	20.00	16.00–27.00	
**Educational level**								0.003 ‡**
Primary school	41	24.07	4.735	15	32	24.00	22.00–27.00	
High school	88	20.82	6.164	11	33	20.00	15.25–26.50	
University	68	20.59	6.228	12	34	19.50	15.25–25.00	
**Marital status**								0.007 ‡**
Married	128	20.95	6.419	11	34	21.00	15.00–26.00	
Unmarried	45	21.87	5.413	14	32	21.00	18.00–25.00	
Widowed	9	18.56	2.128	17	22	17.00	17.00–20.50	
Divorced	15	25.80	3.877	20	30	27.00	22.00–29.00	
**Alcohol consumption**								0.542 †
Yes	47	21.91	6.331	12	32	22.00	17.00–27.00	
No	150	21.26	5.969	11	34	21.00	16.75–26.00	
**Tobacco use**								0.844 †
Yes	39	21.13	6.783	11	32	21.00	14.00–27.00	
No	158	21.49	5.873	11	34	21.00	17.00–26.00	
**Chronic diseases**								0.584 †
Yes	97	21.60	5.662	12	33	22.00	17.00–27.00	
No	100	21.24	6.423	11	34	21.00	16.50–25.00	
**Chronic medication**								0.211 †
Yes	109	21.81	5.289	12	33	22.00	17.00–26.00	
No	88	20.93	6.873	11	34	20.50	15.00–26.00	
**Previous surgery**								0.614 †
Yes	143	21.27	6.129	11	34	21.00	15.00–26.00	
No	54	21.81	5.863	11	31	21.50	18.00–27.00	
**History of cancer**								0.002 †**
Yes	20	25.30	4.414	17	30	26.00	24.00–29.75	
No	177	20.98	6.059	11	34	21.00	16.00–25.00	
**Type of surgery**								0.003 ‡**
Orthopedics	47	24.26	5.825	13	34	23.00	20.00–30.00	
Neurosurgery	41	21.27	5.422	11	31	20.00	18.00–24.00	
General surgery	56	20.77	6.830	11	32	20.50	14.00–27.00	
ENT surgery	53	19.70	5.010	12	33	20.00	15.00–22.50	

Notes: † Mann–Whitney U test; ‡ Kruskal–Wallis test; ** *p* < 0.01 (statistically significant).

**Table 4 medsci-13-00181-t004:** Tests of between-subject effects on preoperatory stress scores.

Source	Type I Sum of Squares	df	Mean Square	F	Sig.	Partial Eta Squared	Effect Interpretation
Type of surgery	559.731	3	186.577	14.278	<0.001 **	0.224	High
Sex	1081.645	1	1081.645	82.773	<0.001 **	0.359	High
Educational level	394.624	2	197.312	15.099	<0.001 **	0.169	High
Marital status	173.017	3	57.672	4.413	0.005 **	0.082	Moderate
History of cancer	415.606	1	415.606	31.804	<0.001 **	0.177	High
Type of surgery × Sex	286.809	3	95.603	7.316	<0.001 **	0.129	Moderate
Type of surgery × Educational level	542.103	6	90.351	6.914	<0.001 **	0.219	High
Type of surgery × Marital status	363.262	7	51.895	3.971	<0.001 **	0.158	Moderate
Type of surgery × History of cancer	149.052	2	74.526	5.703	0.004 **	0.072	Moderate
Sex × Educational level	162.899	2	81.450	6.233	0.003 **	0.078	Moderate
Educational level × Marital status	251.631	4	62.908	4.814	0.001 **	0.115	Moderate
Educational level × History of cancer	451.934	2	225.967	17.292	<0.001 **	0.189	High
Type of surgery × Sex × Educational level	143.869	4	35.967	2.752	0.030 *	0.069	Moderate
Type of surgery × Sex × Marital status	68.112	2	34.056	2.606	0.077 +	0.034	Low
Type of surgery × Educational level × Marital status	138.192	3	46.064	3.525	0.017 *	0.067	Moderate

Notes: * *p* < 0.05; ** *p* < 0.01; + trend toward significance. Effect interpretation according to Cohen’s guidelines (Partial Eta Squared: 0.01–0.059 = low; 0.06–0.139 = moderate; ≥0.14 = high).

## Data Availability

The data presented in this study are available on request from the corresponding author. The data have been anonymized to protect patient privacy.

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
