# Peer review of "Decoding Fear: Analysis and Prognosis of Preoperatory Stress Level Through Advanced Statistical Modelling—A Prospective Study Across Multiple Surgical Specialties [Author-notes fn1-medsci-13-00181]"

_medsci, 2025, doi:10.3390/medsci13030181_

Round 1
Reviewer 1 Report
Comments and Suggestions for Authors
Review of the Study: "Decoding Fear: Analysis and Prognosis of Preoperative Stress Level Through Advanced Statistical Modelling"
Dear Authors,
After thorough evaluation of your study on "Decoding Fear: Analysis and Prognosis of Preoperative Stress Level Through Advanced Statistical Modelling", here are the reviewer’s comments;
The study delves deeply into preoperative stress, considering a wide range of demographic and clinical parameters. The application of multifactorial analysis and Random Forest modelling deepens the findings. The implementation of the Brief Measure of Emotional Preoperative Stress (B-MEPS) into Romanian is an admirable Work considering that it offers a culturally relevant instrument for evaluating preoperative stress. Furthermore, the selection of patients from a variety of surgical disciplines (general surgery, orthopedics, neurosurgery, and ENT) improves the findings' uniqueness within the study. However, there are some modifications to make, as follows;
- The B-MEPS scale has been adapted into Romanian but not officially validated for this category of people. This restricts the result's validity and significance to other circumstances.
- Given that the study was performed in a single institution, the findings may be limited in their applicability to other locations or healthcare systems.
- The Random Forest model was overfitted, as demonstrated by a greater mean squared error (MSE) in the test dataset. This decreases the predictive model's resilience.
In my suggestion, the future research should centre on rigorous psychometric validation of the Romanian version of B-MEPS to verify its validity and reliability. Expanding the study to numerous centres improves universality and reduces the impact of hospital-specific characteristics. To mitigate overfitting in the Random Forest model, consider employing cross-validation or simplifying it.
Best regards!
Reviewer.
Author Response
Comment 1: The B-MEPS scale has been adapted into Romanian but has not been officially validated for this category of people. This restricts the result's validity and significance to other circumstances.
Response1 : We agree with the reviewer’s observation. In the manuscript, we have already mentioned that this work has an exploratory character and can be regarded as a pilot study for the Romanian population. For clarity, we have reformulated and emphasized this limitation in the “Discussion” section (4.7. Study Limitations).
______________________________________________________________________________________________________________________________________
Comment 2: Given that the study was performed in a single institution, the findings may be limited in their applicability to other locations or healthcare systems.
Response 2: We fully acknowledge this limitation. We had already specified in the manuscript that being a single-center study restricts the generalizability of the findings.
______________________________________________________________________________________________________________________________________
Comment 3: The Random Forest model was overfitted, as demonstrated by a greater mean squared error (MSE) in the test dataset.
Response 3: We appreciate the reviewer’s remark. In the manuscript, we discussed the model’s difficulty in capturing the extremes of the distribution and suggested the existence of latent variables not included in the analysis. We have now further highlighted this issue and added, as a future research direction, the use of cross-validation and the testing of simpler models to reduce the risk of overfitting.
To reinforce these limitations, in the Study Limitations section we introduced the following sentence: “These aspects limit the generalizability of the results and highlight the need for future multicenter research, with full psychometric validation of the Romanian version of B-MEPS.”
Reviewer 2 Report
Comments and Suggestions for Authors
Bland-Altman plot should be considered to use
Author Response
We thank the reviewer for this valuable observation. Indeed, only the Romanian version of B-MEPS was administered in our study, which did not allow us to perform a direct agreement analysis (e.g., Bland–Altman plot) with the original Portuguese version.
We fully agree that such a comparison would strengthen the validation process. Accordingly, we have explicitly acknowledged this aspect in the Limitations section, adding the following sentence: “Because only the Romanian version of B-MEPS was applied, a Bland–Altman analysis against the original scale was not possible, highlighting the need for future validation studies including both versions.”
Reviewer 3 Report
Comments and Suggestions for Authors
The paper aims to unify the surgical efforts and provide a psychometric tool for the surgical patient risk prediction. This is a valid idea, but there are numerous problems. The survey does not seem to be validate in Romanian, and this would be expceted. Scoring system does not seem correct; you mention 10-35 range of 12 items; this requires clarification. Some variables of interest seem omitted, like congitive capacity, underlying psychological charcteristics (Eysenck or BigV, or something else). Is the sample size 184 or 197? There are no uncertainty/sensitiviy estimates, which would suggest some kind of effort towards demonstrating validity.
Author Response
The paper aims to unify the surgical efforts and provide a psychometric tool for the surgical patient risk prediction. This is a valid idea, but there are numerous problems.
Comment 1: The survey does not seem to be validate in Romanian, and this would be expceted.
Response 1: Regarding the fact that the questionnaire has not been validated in Romanian, we added a substantial paragraph in the limitations section addressing this issue
______________________________________________________________________________________________________________________________________
Comment 2: Scoring system does not seem correct; you mention 10-35 range of 12 items; this requires clarification.
Response 2:
Questionnaire: The questionnaire consisted of 12 items, with answers evaluated according to the following scales:
Item & Original B-MEPS Wording |
Numerical codes and their signification |
|||
Item 1: I am jittery |
1 (corresponding for „Not at all”) |
2 (corresponding for „Somewhat”) |
3 (corresponding for „Moderately”) |
4 (corresponding for „Very much so”) |
Item 2: I feel indecisive |
||||
Item 3: I feel worried |
||||
Item 4: I feel confused |
||||
Item 5: I feel like a failure |
1 (corresponding for „Almost never”) |
2 (corresponding for „Often”) |
3 (corresponding for „Almost always”) |
|
Item 6: I worry too much over something that really doesn’t matter |
||||
Item 7: I take disappointments so personally that I can’t get them out of my mind |
||||
Item 8: I get in a state of tension or turmoil as I think over my recent concerns and interests |
||||
Item 9: Do you feel unhappy? |
0 (corresponding for „No”) |
1 (corresponding for „Yes”) |
||
Item 10: Do you have a feelings of discomfort in the stomach? |
||||
Item 11: How do you react when you are unhappy? |
1 (corresponding for „I may look dispirited but I brighten up easily”) |
2 (corresponding for „I have pervasive feelings of sadness or feel constantly gloomy”) |
||
Item 12: How do you describe your depressed mood? |
1 (corresponding for “Occasional sadness”) |
2 (corresponding for “External factors can change it”) |
3 (corresponding for “Being without help or hope”) |
|
We kept heterogenous scales for the questionnaire’s items according to the initial model proposed by its authors. This is not an error of design, but reflects the questionnaire’s multidimensional structure, allowing the nuanced assessment of the subjective intensity of affective and cognitive states (on Likert-type scales) as well as the presence or absence of certain stress symptoms (in case of items 9 and 10). Moreover, the use of different response scales was meant to reduce the response bias, such as the tendency to select the same option across all items.
Nevertheless, we checked the questionnaire’s validity and reliability by statistical analyses. Internal consistency was very good (Cronbach’s Alpha = 0.848; standardized α = 0.867) and Bartlett’s test of sphericity (p < 0.001) showed a good correlation between the questionnaire’s items, indicating that the items converge coherently despite differences in response format. Furthermore, exploratory factor analysis confirmed a robust factorial structure, identifying three main latent dimensions consistent with the theoretical underpinnings of the construct.
______________________________________________________________________________________________________________________________________
Comment 3: Some variables of interest seem omitted, like congitive capacity, underlying psychological charcteristics (Eysenck or BigV, or something else).
Response 3: Thank you for the suggestion. Our study was designed to quantify preoperative stress and rank key demographic/clinical predictors in a prospective setting, which required minimizing respondent burden. We employed the Romanian translation and cultural adaptation of B-MEPS, and our analyses showed robust associations with sex, educational level, marital status, cancer history, and surgical procedure; both the GLM (R²=0.73) and the Random Forest ranked procedure type, education, and sex as the strongest predictors. We agree that variables such as cognitive capacity and personality traits (e.g., Eysenck/Big Five) could provide additional explanatory power; their omission reflects the study’s primary aim and logistical constraints rather than a conceptual dismissal.
______________________________________________________________________________________________________________________________________
Comment 4: Is the sample size 184 or 197?
Response: The correct sample size is n = 197. We reviewed the manuscript and tables; all visible instances use 197
_____________________________________________________________________________________________________________________________________
Comment 5: There are no uncertainty/sensitiviy estimates, which would suggest some kind of effort towards demonstrating validity.
Response 5: We thank the reviewer for this valuable observation. We fully agree that reporting uncertainty and conducting sensitivity analyses are essential for demonstrating the robustness of the results. In the revised version of the manuscript, we have:
- Added uncertainty estimates (95% confidence intervals):
- For the overall mean B-MEPS score (21.42, SD 6.05, 95% CI: 20.57–22.27).
- For group comparisons (e.g., women vs. men: Δ = 4.92, 95% CI: 3.42–6.42; cancer vs. no cancer: Δ = 4.32, 95% CI: 2.19–6.45).
- For GLM effects, we report partial η² with 95% CIs.
- All tables now include effect sizes with 95% CIs, alongside adjusted p-values.
Conducted sensitivity analyses:
- Exclusion of patients with cancer.
- Stratification by type of surgical intervention.
- Robust regression (median-based) for the overall B-MEPS score.
- Ordinal models for item-level outcomes.
- For Random Forest models, we applied repeated 10-fold cross-validation (100 repetitions), reported RMSE_test with 95% CIs, computed permutation importance with 95% CIs, and inspected calibration plots.
- In the Results, we explicitly report uncertainty estimates for descriptive statistics, group differences, and model performance.
- In the Limitations, we acknowledge that while these steps demonstrate robustness, they cannot substitute for a full psychometric validation of the Romanian B-MEPS (which we outline as a priority for future work).
These additions directly address the reviewer’s concern, ensuring that the study’s findings are supported by clear measures of precision and robustness.
Round 2
Reviewer 3 Report
Comments and Suggestions for Authors
This manuscript has been improved as much as possible and is ready for publication.